# Detection of Ovine or Bovine Milk Components in Commercial Camel Milk Powder Using a PCR-Based Method

**DOI:** 10.3390/molecules27093017

**Published:** 2022-05-07

**Authors:** Xiaoyun Wu, Qin Na, Shiqi Hao, Rimutu Ji, Liang Ming

**Affiliations:** College of Food Science and Engineering, Inner Mongolia Agricultural University, Hohhot 010018, China; wuxy_imau@163.com (X.W.); naqin_imau@163.com (Q.N.); haosq_imau@163.com (S.H.); yeluotuo1999@vip.163.com (R.J.)

**Keywords:** camel milk powder, adulteration detection, PCR-based method, qualitative and quantitative

## Abstract

Food ingredient adulteration, especially the adulteration of milk and dairy products, is one of the important issues of food safety. The large price difference between camel milk powder, ovine, and bovine milk powder may be an incentive for the incorporation of ovine and bovine derived foods in camel milk products. This study evaluated the use of ordinary PCR and real-time PCR for the detection of camel milk powder adulteration based on the presence of ovine and bovine milk components. DNA was extracted from camel, ovine, and bovine milk powder using a deep-processed product column DNA extraction kit. The quality of the extracted DNA was detected by amplifying the target sequence from the mitochondrial Cytb gene, and the extracted DNA was used for the identification of milk powder based on PCR analysis. In addition, PCR-based methods (both ordinary PCR and real-time PCR) were used to detect laboratory adulteration models of milk powder using primers targeting mitochondrial genes. The results show that the ordinary PCR method had better sensitivity and could qualitatively detect ovine and bovine milk components in the range of 1% to 100% in camel milk powder. The commercial camel milk powder was used to verify the practicability of this method. The real-time PCR normalization system has a good exponential correlation (R^2^ = 0.9822 and 0.9923) between ovine or bovine content and Ct ratio (specific/internal reference gene) and allows for the quantitative determination of ovine or bovine milk contents in adulterated camel milk powder samples. Accuracy was effectively validated using simulated adulterated samples, with recoveries ranging from 80% to 110% with a coefficient of variation of less than 7%, exhibiting sufficient parameters of trueness. The ordinary PCR qualitative detection and real-time PCR quantitative detection method established in this study proved to be a specific, sensitive, and effective technology, which is expected to be used for market detection.

## 1. Introduction

As a special milk source, camel milk contains a large amount of immune active proteins such as lysozyme, lactoferrin, lactoperoxidase, immunoglobulin, vitamin C, and insulin and is also rich in calcium, potassium, iron, and other minerals [1,2]. Camel milk is different from other ruminant milks in that it has high levels of odd- and branched-chain fatty acid and low ratios of n-6 to n-3 polyunsaturated fatty acid, and is therefore considered a potential functional food for a balanced human diet [3]. Hinz et al. compared the protein composition differences of milk of different animals such as camel, buffalo, goat, cow, and horse by a proteomic method and found that camel milk is similar to human milk in the absence of the allergenic milk protein β-lactoglobulin [4]. Camel milk is nutritionally suitable for human consumption and is a promising alternative to cow’s milk in infant formula [5]. In addition, there are a number of studies showing that camel milk also has antioxidant, hepatoprotective, anti-inflammatory, and antibacterial properties as well as other effects and can also be used in adjuvant treatment of cancer, diabetes, kidney disease, liver disease and intestinal inflammation [6,7,8]. With the continuous discovery of camel milk’s special nutritional components and efficacy, consumer demand has also increased, and the substitution or adulteration of camel milk powder with ovine or bovine milk powder has often occurred. The reason is that, while ovine and bovine milk have a similar color and taste to camel milk, their price is lower, and since ordinary consumers cannot distinguish the differences, they are often used by illegal traders to adulterate camel milk [9]. To avoid fraudulent substitution of camel milk powder with ovine or bovine milk powder, it is necessary to develop analytics that can detect such fraud and protect consumers from misleading labeling.

Existing analytical methods for authenticating milk and dairy products include protein-based methods and DNA-based methods. Previous studies have developed methods based on capillary electrophoresis (CE) and enzyme-linked immunosorbent assay (ELISA) using bovine α-lactalbumin and β-casein as adulteration markers [10,11]. The presence of hydrolyzed vegetable proteins in milk was detected by nano-high performance liquid chromatography (HPLC)-tandem mass spectrometry (LC-MS/MS), and the corresponding accuracy would be validated by sodium dodecyl sulfate-polyacrylamide gel electrophoresis (SDS-PAGE) [12]. Such methods have been successfully applied to the identification of dairy products and have played an important role in the food industry. However, targeting proteins as target analytes may not be practical for high-temperature and high-pressure processed dairy products because these proteins are less thermally stable, leading to changes in antigenicity and electrophoretic mobility [13]. In recent years, DNA molecules have attracted attention due to their good thermal stability and high specificity, and have been used in polymerase-linked reactions [14]. DNA-based methods are currently widely used, especially ordinary PCR and real-time PCR techniques. Maudet et al. developed a target gene targeting the control region sequence of bovine mitochondrial DNA (mtDNA) and used ordinary PCR technology to detect milk DNA in goat cheese, which proved to be very sensitive, with a detection limit lower than 0.1% of cow’s milk [15]. Pinto et al. amplified the bovine mitochondrial cytochrome b gene using ordinary PCR technology, and detected the milk components in 30 Italian buffalo milk cheeses, and the results showed that 22/30 samples had milk genomes, confirming the feasibility of this method [16]. However, ordinary PCR cannot be used as an accurate quantitative tool for dairy adulteration, thus real-time PCR is often used as a quantitative tool for dairy product certification, and this method has been used for detecting the adulteration of camel milk and its products. Wu et al. established a real-time PCR detection method based on the Cytb gene, which can detect the camel DNA content in 0.01% bovine DNA with high sensitivity [17]. Wajahat et al. evaluated real-time PCR for the detection of cow and goat milk incorporation in camel milk and found a detection limit of 0.001–0.002% for camel DNA templates [18]. PCR-based methods help to identify components of food matrices through specific gene amplification and have potential in addressing food adulteration [19].

In order to determine whether the camel milk powder samples were adulterated and the proportion of adulteration, this study used ordinary PCR and real-time PCR to qualitatively and quantitatively detect ovine and bovine milk in adulterated commercial camel milk powder. Species identification was carried out by amplifying the target sequences of mitochondrial genes by ordinary PCR, and then quantitatively detecting the adulteration ratio of ovine or bovine milk in camel milk powder by real-time PCR. Furthermore, the establishment of a standard curve based on ovine or bovine milk powder content (ovine or bovine milk powder/total milk) versus Ct ratio (specific/reference gene) corrected the deviation of DNA extraction and the difference of PCR amplification efficiency between the species-specific primers and reference primers [20]. This study established a simple, reliable method for qualitative and quantitative detection of ovine and bovine milk components in camel milk powder, and also provided a reference value for the detection of adulteration in milk and dairy products.

## 2. Materials and Methods

### 2.1. Materials

Ezup Food Genomic DNA Extraction Kit, Taq Plus DNA Polymerase, PCR buffer (10×), dNTP mixture (10 mM), SYBR Green Fast qPCR Master Mix (2×), and DNase/RNase-free water were from Sangon Biotechnology Co. Ltd. (Shanghai, China). CHCl_3_ was purchased from Damao Chemical Reagent Factory (Tianjin, China). Whole milk powder was obtained from a supermarket in Hohhot (Inner Mongolia, China) and their origin species were authenticated by subjecting the samples to species typing using a PCR-based approach.

### 2.2. DNA Extraction and Quality Evaluation

#### 2.2.1. DNA Extraction Method of Milk Powder

A sample of approximately 650 mg was taken and dissolved in 1 mL CMO Buffer. The mixture was vortexed and 20 μL Proteinase K was added, followed by incubation at 65 °C for 60 min and then centrifugation at 12,000 rpm for 10 min. The supernatant was collected, and 400 μL of CHCl_3_ was added. The obtained sample was centrifuged at 12,000 rpm for 10 min, and the supernatant was collected and the same volume of GMP Buffer and 350 μL DRL Buffer were added. The mixture was collected in an EZ-10 Spin Column, and the column was centrifuged at 12,000 rpm for 5 min. The EZ-10 Spin Column was first washed with 500 μL Wash Solution using the same aforementioned centrifugation speed and rewashed using Wash Solution at 10,000 rpm centrifugation speed for 2 min. The EZ-10 Spin Column was then placed in a clean 1.5 mL Eppendorf tube and the DNA was eluted using 50 μL TE Buffer (pH 8.0) with centrifugation at 12,000 rpm for 2 min, and the DNA solution was collected.

#### 2.2.2. Evaluation of DNA Quality

The concentration and purity of total DNA were determined through absorbance readings at 260 and 280 nm using the Nanodrop ND-ONE Spectrophotometer (Thermo Fisher Scientific, Waltham, MA). In addition, the bovine, ovine and camel Cytb genes (Table 1) were amplified by PCR to verify the presence of amplifiable mitochondrial DNA in all isolates. The PCR reactions of Cytb were performed under the following thermal cycling conditions: initial denaturation at 95 °C for 5 min; followed by 10 cycles of 94 °C for 30 s, 63 °C (with a reduction by 0.5 °C for each successive cycle) annealing for 30 s, 72 °C for 30 s; followed by 30 cycles of 95 °C for 30 s, 58 °C annealing for 30 s, 72 °C for 30 s; and a final extension at 72 °C for 10 min. All of the Cytb amplification reactions were carried out in a mixture (20 μL) comprising 1 μL of dNTP mixture (10 mM), 2.5 μL of PCR buffer (10×), 1 μL of each primer (5 μM), and 1 μL of the template DNA and DNase/RNase-free water was added to a final reaction volume of 20 μL. The PCR products were detected using 1% agarose gel electrophoresis at 150 V for 15 min and photographed under UV light.

### 2.3. Qualitative Detection of Bovine Milk and Ovine Milk in Camel Milk Powders

#### 2.3.1. Validation of Primers

In order to verify the specificity of the primers used in this experiment and the purity of milk powder, DNA was extracted from whole camel milk powder, ovine milk powder and bovine milk powder, and the extracted DNA template was amplified with the specific primers for the camel, ovine, and bovine Cytb genes in Table 1 to ensure the specificity of the primers used in this experiment and the purity of the milk powder.

#### 2.3.2. Qualitative Detection of Camel Milk Powder

Ovine and bovine specific primer Cytb (Table 1) was selected for detection of ovine and bovine milk in camel milk powder by amplifying fragments corresponding to the mitochondrial Cytb gene. For the spiked model, admixtures compromising 0.5%, 1%, 3%, 5%, 10%, 30%, 50%, and 100% (wt/wt) ovine milk powder or bovine milk powder in camel milk powder were prepared, with preparation of each sample repeated 3 times. The DNA isolated from these mixtures was analyzed according to a method we developed for PCR detection. The PCR products were subsequently electrophoresed on a 1% agarose gel at 150 V for 15 min and photographed for analysis under UV light. 

#### 2.3.3. Detection of Commercial Camel Milk Powder

The applicability of the PCR qualitative detection method in terms of authentication was verified. DNA was extracted from ten samples of camel milk powder of five different brands and amplified with the established method to characterize the purity of these camel milk powder samples. 

### 2.4. Qualitative Detection of Bovine Milk and Ovine Milk in Camel Milk Powders

#### 2.4.1. Validation of Primers

Ovine mitochondrial 12S rRNA gene and bovine mitochondrial Cytb gene were used as specific genes, and ruminant mitochondrial 16S rRNA gene was used as an internal reference gene (Table 1). The specificity and universality of the primers were verified by using specific primers and reference primers to perform amplification on milk powder DNA. The PCR was carried out under the following thermal cycling conditions: 95 °C for 3 min followed by 45 cycles of 95 °C for 5 s, and 60 °C annealing for 30 s to collect the fluorescence signal at the end of each cycle. A 10 μL singleplex reaction was prepared comprising 5 μL of SYBR Green qPCR Master Mix (2×), 0.2 μL of each primer (10 μM), 1 μL of the template DNA and DNase/RNase-free water was added to a final reaction volume of 10 μL. Judgment of specific primer amplification results: if Ct value < 30, the results are valid; if Ct value ≥30, the result is discarded. NTC is the blank control. 

#### 2.4.2. Preparation of Quantitative Calibration Curve

To quantify the adulterated content of ovine milk and bovine milk in camel milk powder, specific primers and internal reference primers were used to amplify the DNA of camel milk powder mixture containing 5%, 10%, 30%, 50%, 80%, and 100% ovine milk powder or bovine milk powder, with amplification of each sample repeated 3 times. The obtained Ct ratio (specific primer Ct value/internal reference primer Ct value) was used as the *y*-axis, and the percentage of ovine milk powder or bovine milk powder was used as the *x*-axis to fit an exponential regression curve. The total amount of mixed samples was represented by a ruminant mitochondrial 16S rRNA gene, and the amount of ovine or bovine samples was represented by ovine mitochondrial 12S rRNA gene and bovine mitochondrial Cytb gene. Therefore, the percentage content of ovine milk powder or bovine milk powder by Ct ratio (ovine or bovine specific primers Ct value/internal primers Ct value) can be more appropriately determined using mixed samples rather than sheep powder and the powder percentage based on standard curve fitting, and the ovine or bovine powder percentage can be deduced by calculation of the Ct ratio [24]. 

#### 2.4.3. Analysis of Adulterated Simulations

To evaluate the accuracy and precision of the normalized system, adulterated simulations were analyzed by mixing ovine or bovine milk into camel milk at ratios of 30%, 50% and 80% in triplicate. The content of camel milk in the simulation samples was obtained using the calibration model. The recoveries, standard deviation (SD), and coefficient of variation (CV) were calculated to verify the accuracy of the quantitative method.

## 3. Results and Discussion

### 3.1. Quality of DNA Extracted from Milk Powders

#### 3.1.1. DNA Yield and Purity

The concentration and purity of the DNA template determine the success of PCR detection, so the quality of the DNA template is particularly important for subsequent tests [25]. In relation to milk powder, the DNA concentrations were 50–140 ng/µL, and the purity ratios (A260/A280; absorbance at 260 and 280nm) ranged from 1.69 to 1.94, which is close to the range of 1.7 to 1.9 for pure DNA samples [26]. There were slight differences in the DNA concentration and purity of the different milk samples.

#### 3.1.2. DNA Integrity

The integrity of the DNA was reflected by amplifying the mitochondrial Cytb gene fragments of interest [27]. Representative agarose gel electrophoresis analysis (Figure 1) showed positive results for the integrity of mitochondrial DNA extracted from camel, ovine, and bovine milk powders, and the resulting bands were clear and consistent, indicating the high integrity of samples with no smearing or diffusion. This indicates that a sufficient enough amount of high-quality DNA was isolated through the method we developed in the present study, which is expected to be useful for species identification. Primer–dimers are present in all amplified sequences. However, any DNA (background or specificity), especially double-stranded DNA, promotes primer–dimer formation even for primers that would not form these products in the absence of DNA [28]. Soares and Deng et al. concluded that, although analysis of PCR amplification products indicated the presence of primer dimers, they did not qualitatively affect the experimental results [29,30].

### 3.2. Results of Qualitative Detection

#### 3.2.1. Validation of Primers

Sample DNA templates and primers are critical components of any PCR analysis, as they are the primary determinants of its specificity, sensitivity, and robustness [31]. Therefore, the purity of milk powder samples and the specificity of primers have a great influence on the results of subsequent experiments, which are related to the success or failure of the experiments [32]. The PCR amplification results for camel, ovine, and bovine milk powder DNA showed that only the expected products amplified by specific primers for the corresponding sample showed positive amplification, and the remaining samples did not produce nontarget bands (Figure 2). This result demonstrates that the camel, ovine, and bovine milk powders are from pure breeds, and that the camel, ovine, and bovine primers used in this study are specific. Thus, the milk powder samples and the designed specific primers selected in this study can be used in subsequent PCR experiments.

#### 3.2.2. Qualitative Detection of Camel Milk Powder

The ordinary PCR assay was optimized for discriminating ovine milk or bovine milk in camel milk powder matrices for detecting adulteration of camel milk powder with ovine or bovine milk powder. The ordinary PCR qualitative detection results showed that the method could clearly detect the incorporated ovine milk (Figure 3) or bovine milk (Figure 4) components in camel milk powder, and the minimum LOD was 1% (wt/wt). In addition, the fluorescence band intensities we obtained showed a trend of being enhanced with an increasing degree of adulteration. Cheng et al. [33] used ordinary PCR to detect the components of bovine milk powder mixed in goat milk powder, and we achieved a comparable sensitivity with our test. The PCR tests used in this screening were therefore demonstrated to be very sensitive, specific, and reproducible.

#### 3.2.3. Detection of Commercial Camel Milk Powder

We used the qualitative detection method established in this study to examine 10 commercial camel milk powders. The results of this test show that samples 1, 2, 3, 5, 9, and 10 only have positive amplification in the lanes amplified with camel-specific primers, indicating that these six samples are pure camel milk powders that do not contain bovine and ovine milk components. The lanes amplified by ovine-specific primers of samples 4, 7, and 8 also showed positive amplification, showing that these three samples were camel milk powder with different levels of ovine milk. The two lanes of sample 6 amplified with ovine- and bovine-specific primers showed positive amplification, indicating that this sample had camel milk powder with both ovine and bovine milk components (Figure 5). The DNA in milk powder is derived from somatic cells, and factors such as animal health, lactation period, season, and processing methods can affect the number of somatic cells in milk powder, thereby changing the DNA content [34]. Therefore, the DNA content of different brands of camel milk powder samples varies, resulting in inconsistent brightness of PCR amplification bands. In the future, the DNA extraction method can be further optimized to improve the quality of primers, which can be applied to the adulteration detection of camel milk powder in the market.

### 3.3. Results of Quantitative Detection

#### 3.3.1. Validation of Primers

The specificity and versatility of primers in real-time PCR detection methods have a significant impact on the estimation of research results [35]. The reference primers used in the present study were based on ruminants; hence, their generality was assessed by DNA extracted from camel, ovine, and bovine milk powder. As shown in Figure 6, samples from all species showed amplification with Ct values of 28.28, 24.46, and 30.53, indicating good primer versatility. Due to the different copy numbers in tissues of various species, the Ct values of the internal reference primers for amplification are different [36]. The specificity of ovine- and bovine-specific primers was verified using DNA extracted from ovine and bovine milk powder as a positive template, and nontarget species as negative templates. The results in Figure 7A, B show that, within 30 cycles, significant amplification signals were obtained from only ovine and bovine milk powders, and no amplification was observed for camel milk powder. Meanwhile, the NTC did not yield any amplification signals. Thus, this result demonstrates that the reference primers were universal and that the ovine and bovine primers were highly specific to the target species.

#### 3.3.2. Preparation of Quantitative Calibration Curve

Different concentrations (5–100%) of ovine or bovine milk were used to spike camel milk powder samples, which were then subject to detection by specific primers. Figure 8A shows that the amplified Ct values of 5%, 10%, 30%, 50%, 80%, and 100% ovine milk powder were 25.35, 24.41, 23.25, 22.23, 20.75, and 19.42, respectively. As shown in Figure 8B, the amplified Ct values of 5%, 10%, 30%, 50%, 80%, and 100% bovine milk powder were 28.25, 27.78, 26.38, 25.21, 23.58, and 22.83, respectively. All amplified Ct values were less than 30, indicating that the experimental data were reliable. Due to possible differences in the extraction or amplification efficiency between the ovine or bovine species-specific genes and reference genes, the linear relationship between Ct ratio (specific/reference genes) and ovine or bovine milk powder content (ovine or bovine/total milk powder) can reduce the error and calculate ovine or bovine milk content [20]. Figure 9A,B show that the ratio of Ct to ovine or bovine milk powder content exhibits a good exponential response when the ovine or bovine milk powder content is in the range of 5–100%, with a correlation coefficient (R^2^) of 0.9822 and 0.9923, calibration model y = 0.925 × 10^−0.002x^ and y = 0.9976 × 10^−0.003x^, respectively. The obtained data for coefficient of determination (R^2^) greater than 0.98 of the standard curve may be acceptable according to the literature [37,38]. At the same time, it is also shown that the quantitative detection range for ovine or bovine milk components is from 5% to 100% in adulterated camel milk powder.

#### 3.3.3. Analysis of Adulterated Simulations

Using DNA from 30%, 50%, and 80% goat milk or cow milk mixed with camel milk powder mixed samples as adulteration models, PCR amplification was performed to analyze measured and actual values. All assay results are presented in Table 2 and Table 3, indicating that the recoveries of all simulated samples were between 80% and 110%, and the CV values were all less than 7% of the acceptable range [39], which confirms the accuracy and precision of the standardized system. 

## 4. Conclusions

As the demand for camel milk and its products has increased, incidents of adulteration of camel milk and its products have also received more attention. DNA-based PCR detection methods have high sensitivity and specificity and are widely used in the detection of adulteration of milk and dairy products. Extraction of DNA is one of the most important factors that can affect the successful implementation of PCR-based methods. Therefore, we used a column kit method to extract high-quality DNA from milk powder for PCR research on milk powder. Ordinary PCR was used for qualitative detection of adulterated ovine and bovine milk components in commercial camel milk powder, with a detection range of 1% to 100%; real-time PCR was used for quantitative detection, with a detection range of 5% to 100%. In addition, the authenticity and practicability of the ordinary PCR qualitative detection method were evaluated using 10 commercial camel milk powder samples. Using adulteration simulations to evaluate the accuracy and precision of a standardized system for real-time PCR quantitative detection, recoveries ranged from 80% to 110% and CV values were less than 7%. The method established in this study is simple, low cost, and has high sensitivity based on ordinary PCR qualitative detection and real-time PCR quantitative detection of mitochondrial genes, which provides technical support for the adulteration detection of camel milk powder in the market. At the same time, it provides a new idea for adulteration detection of other exogenous substances (such as: cereals, soys, etc.) in dairy products.

## Figures and Tables

**Figure 1 molecules-27-03017-f001:**
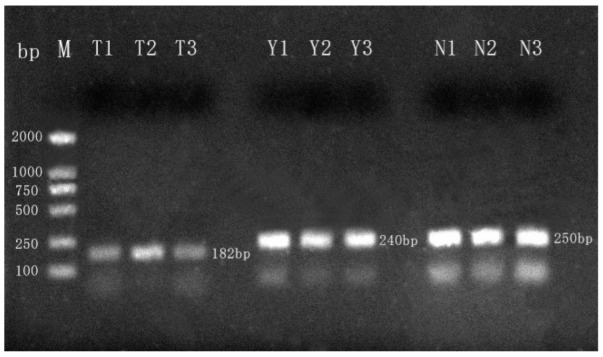
Representative results from agarose gel electrophoresis analysis of PCR products. Lanes T1, T2, T3 = camel Cytb gene amplified camel milk powder DNA template; lanes Y1, Y2, Y3 = ovine Cytb gene amplified ovine milk powder DNA template; lanes N1, N2, N3 = bovine Cytb gene amplified bovine milk powder DNA template; M = molecular weight marker (2000 bp DNA ladder).

**Figure 2 molecules-27-03017-f002:**
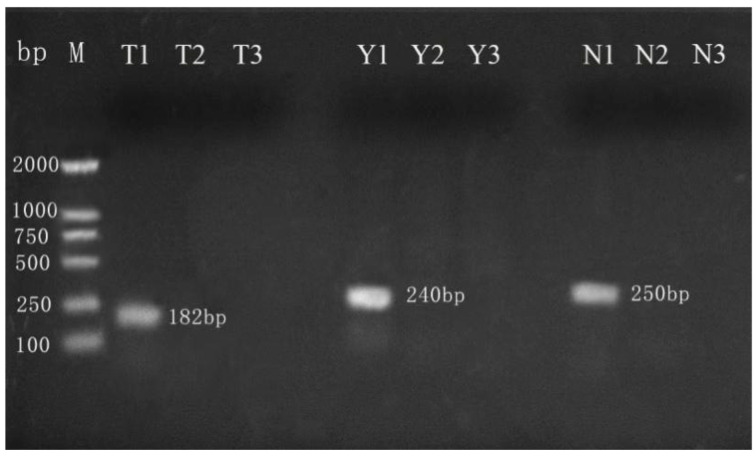
Representative results from agarose gel electrophoresis analysis of PCR validation. Lanes T1, T2, T3 = camel milk powder DNA template amplified with camel, ovine, and bovine Cytb genes, respectively; lanes Y1, Y2, Y3 = ovine milk powder DNA template amplified with camel, ovine, and bovine Cytb genes, respectively; lanes N1, N2, N3 = bovine milk powder DNA template amplified with camel, ovine, and bovine Cytb genes, respectively; M = molecular weight marker (2000 bp DNA ladder).

**Figure 3 molecules-27-03017-f003:**
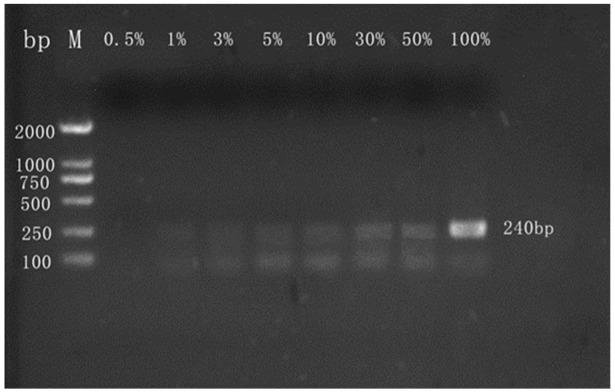
Representative results from agarose gel electrophoresis analysis of PCR products. Lanes 0.5%, 1%, 3%, 5%, 10%, 30%, 50%, and 100% = mixture of camel and ovine milk powders containing 0.5%, 1%, 3%, 5%, 10%, 30%, 50%, and 100% ovine milk composition; Primers were used for the ovine Cytb gene; M = molecular weight marker (2000 bp DNA ladder).

**Figure 4 molecules-27-03017-f004:**
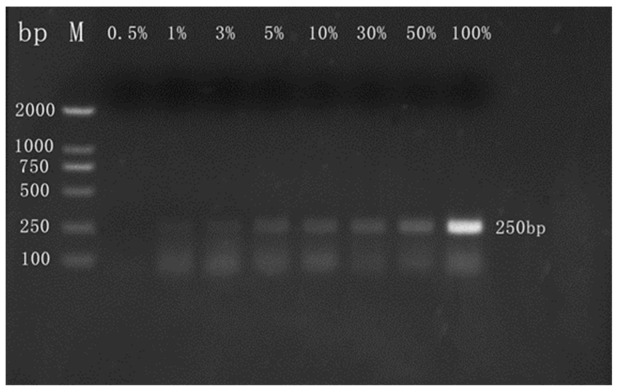
Representative results from agarose gel electrophoresis analysis of PCR detection. Lanes 0.5%, 1%, 3%, 5%, 10%, 30%, 50%, and 100% = mixture of camel and bovine milk powders containing 0.5%, 1%, 3%, 5%, 10%, 30%, 50%, and 100% bovine milk composition; Primers were used for the bovine Cytb gene; M = molecular weight marker (2000 bp DNA ladder).

**Figure 5 molecules-27-03017-f005:**
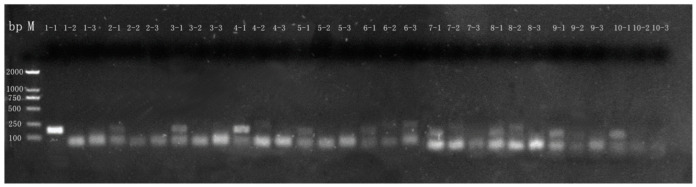
Representative results from agarose gel electrophoresis analysis of PCR applications. Lanes 1–1 = amplification of the DNA template of sample 1 using camel-specific primers; lanes 1–2 = amplification of the DNA template of sample 1 using ovine-specific primers; lanes 1–3 = amplification of the DNA template of sample 1 using bovine-specific primers; M = molecular weight marker (2000 bp DNA ladder).

**Figure 6 molecules-27-03017-f006:**
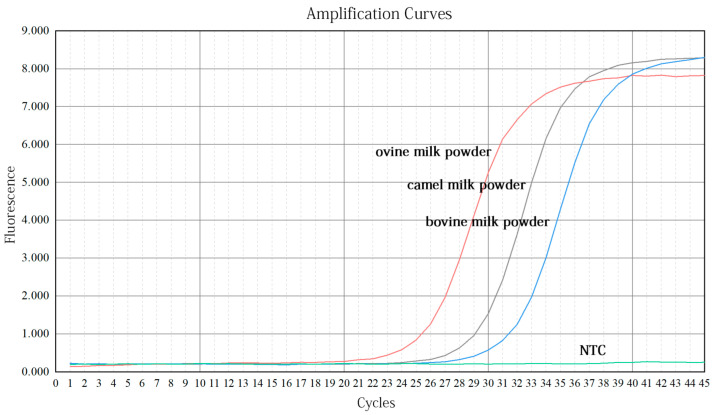
Fluorescence amplification curves of reference primers. *x*-axis = the number of cycles; *y*-axis = the fluorescence intensity.

**Figure 7 molecules-27-03017-f007:**
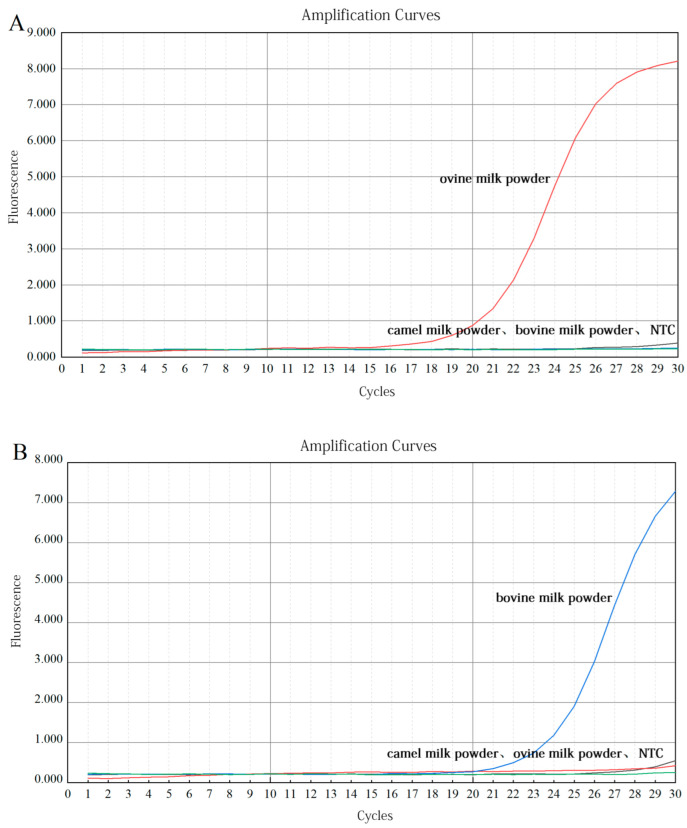
Fluorescence amplification curves of ovine (**A**) or bovine (**B**) specific primers. *x*-axis = the number of cycles; *y*-axis = the fluorescence intensity.

**Figure 8 molecules-27-03017-f008:**
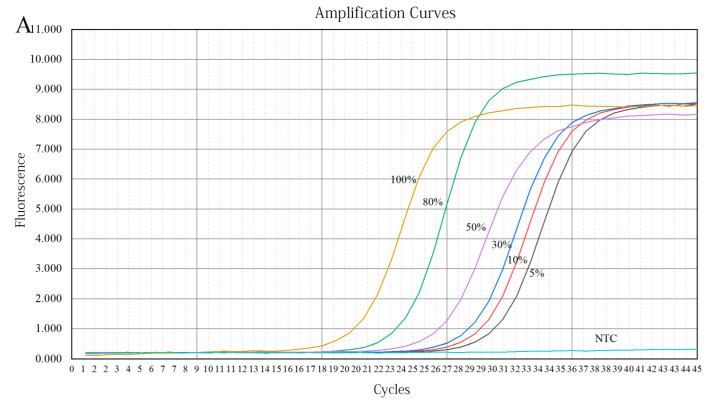
This is a figure. Fluorescence amplification curves of a series of different concentrations of goat milk powder (**A**) or cow milk powder (**B**) in a mixed sample of camel milk powder. *x*-axis = the number of cycles; *y*-axis = the fluorescence intensity.

**Figure 9 molecules-27-03017-f009:**
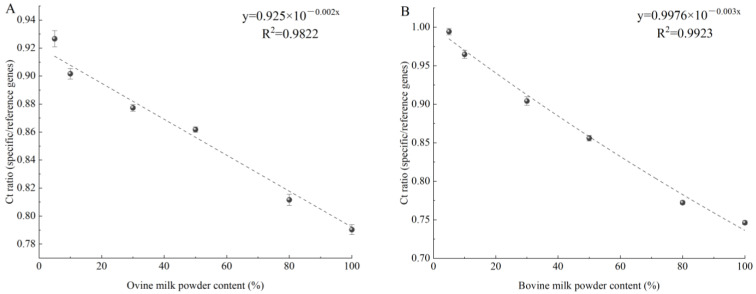
Normalized calibration model used to quantify the content of ovine milk powder (**A**) or bovine milk powder (**B**) in the reference mixture. *y*-axis = Ct ratio (specific/reference gene); *x*-axis = ovine milk powder or bovine milk powder content (ovine milk powder or bovine milk powder/total milk powder).

**Table 1 molecules-27-03017-t001:** Sequences of the primers.

Species	Primer Sequence (5′-3′)	Gene	Amplicons (Bp)	Origin of Primer Sequences
camel	F: CATTATCACGGCTCTAGTGGC	Cytb	182	Sangon Biotechnology Co. Ltd.
	R: CTGGTGAGAATAATACGAGGATAAG			
ovine	F: GAGTAATCCTCCTATTTGCGAC	Cytb	240	Sangon Biotechnology Co. Ltd.
	R: GAACTATGGCGAGGGCTGC			
bovine	F: GTACTATTTGCGCCCAACCTCC	Cytb	250	Sangon Biotechnology Co. Ltd.
	R: AGAACAGGCATTGGCTGAGCA			
ovine	F: CAGCCTTCCTGTTAACTTTCAATAG	12S rRNA	106	Zheng et al. [21]
	R: RGTGCTTGATACCTGCTCCTTTTAG			
bovine	F: CAACAGGAATCTCCTCAGACGTAGA	Cytb	91	Fan et al. [22]
	R: GCTAGAATTAGTAAGAGGGCCCCTAA			
ruminant	F: AAGACGAGAAGACCCTTGGACTTTA	16S rRNA	234–262	Luo et al. [23]
	R: GATTGCGCTGTTATCCCTAGGGTA			

**Table 2 molecules-27-03017-t002:** Data for the detection of simulated adulterated camel milk powder samples according to the actual content of ovine milk powder.

Actual Content (%)	Calculated Content (%)	Mean ± SD (%)	CV (%)	Recovery (%)
30	27.28	26.40 ± 1.45	5.51	88.0
	24.72			
	27.20			
50	41.73	41.84 ± 0.19	0.45	83.68
	42.05			
	41.73			
80	70.82	71.57 ± 2.51	3.50	89.46
	74.36			
	69.52			

**Table 3 molecules-27-03017-t003:** Data for the detection of simulated adulterated into camel milk powder samples according to the actual content of bovine milk powder.

Actual Content (%)	Calculated Content (%)	Mean ± SD (%)	CV (%)	Recovery (%)
30	30.40	32.55 ± 1.96	6.01	108.51
	33.04			
	34.22			
50	49.50	50.96 ± 1.38	2.71	101.93
	51.15			
	52.24			
80	85.05	85.16 ± 1.01	1.19	106.45
	86.22			
	84.21			

## Data Availability

Not applicable.

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
