# Peer review of "Detection of Ovine or Bovine Milk Components in Commercial Camel Milk Powder Using a PCR-Based Method"

_molecules, 2022, doi:10.3390/molecules27093017_

Round 1
Reviewer 1 Report
Comments to authors
The manuscript “Detection of Ovine or Bovine Milk Components in Commercial Camel Milk Powder Using PCR-Based Method” deals with the development of the method of detection of adulteration of different kinds of milk (ovine and bovine) in camel milk. This study will have much impact on the dairy industries to provide the certificate of purity of camel milk. The manuscript is well designed and compiled to provide a robust result that is very much direct and clearly described. I found minor corrections which need attention before the final decision is made.
Specific comments:
- LN 19: Kindly check the superscript of R2
- LN 24-27: kindly rewrite the future impact of the study with better clarity.
- LN 52: kindly cite some references here.
- LN 63-66: Kindly rewrite the aims and objectives of the study in detail.
- Section 2.1.2 needs more discussion. Kindly discuss this in detail by providing some more facts and references.
- Section 2.2.3 also need to be discussed
- The conclusion is too short and not discussed in detail. Kindly discuss the conclusion section by improving the future impact of this study.
- Best wishes to the author for this wonderful research work in the area of diagnostics.
Author Response
Dear Reviewer:
Please refer to the attachment for revision comments. thank you.

Reviewer 2 Report
Paper presents a methodology based on real-time and ordinary PCR which can be used to detect the presence of another type of milk in camel milk powder, the last one being recognized for its therapeutical applications, in special due to its ability to promote general well-being, and body natural defenses. The results show that the ordinary PCR method had better sensitivity and could qualitatively detect ovine and bovine milk components in the range of 1% to 100% in camel milk powder. The paper is well written, the methodologies are clearly described, and the results sustain the final conclusions. For this reason, I recommend publishing this article.
I have only two minor observations which can also be solved by editors:
1) More attention to the mistakes due to typewritten
2) The chapter entitled Materials and Methods must be introduced before the Chapter named Introduction
Author Response

(The authors gave the same response as above.)

Reviewer 3 Report
This manuscript reports about the detection of adulterations of camel milk. However, several aspects need to be reconsidered:
1) As abstracts are often placed away from the main text (e.g., in databases), they need at least some information on the background story.
2) Manuscript needs to be checked with regard to supercript and subscript letters and numbers.
3) "immune active proteins such as lysozyme, lactoferrin, lactoperoxidase, immunoglobulin, vitamin C and insulin and is also rich in calcium, potassium, iron and other minerals" ? Does this differentiate from other milk origins ? I do not think so. Looks more like an advertisment. More descriptions on uniqueness and importance necessary.
4) Introduction is by far very general. It needs more emphasis on what has been already done IN DETAIL in the past and what kind of improvements might be necessary now. Usually, clear hypotheses might help underline a scientific approach. What was expected before the study ? Why this-and-that was selected as being helpful/more helpful (e.g., why not doing MS ?) ? Rationales and development of the methodologies need to be explained far better.
5) Aim is also not clear enough. It is more redundant from the text and very general.
6) However, I miss a discussion at hand of literature. I cannot believe that such (or similar) adulterations have not been analyzed in the past. What is new ? Improvement ? Uniqueness ? Superior ? Development of the topic itself ?
7) Methods are not properly described. Abbreviations not explained. E.g., What is a CMO buffer ? etc.
8) Usually, SI units are "µL" instad of "µl" etc.
9) Quality of Figures is quite bad. Font size often too small...Who should read and understand this ?
Author Response

(The authors gave the same response as above.)

Round 2
Reviewer 3 Report
However, revised version shows lot of improvements, but why is chapter 1 now Material&Methods ? This kind of usual or did I miss a change in the guide for authors ?
Author Response
Dear Reviewer:
Thank you for your suggestion again. Please look at the attachment. If you need to modify it, I hope you will give your suggestion again, and I will definitely modify it carefully.
